# Direct binding of phosphatidylglycerol at specific sites modulates desensitization of a ligand-gated ion channel

Ailing Tong[1], John T Petroff II[1], Fong-Fu Hsu[2], Philipp AM Schmidpeter[3], Crina M Nimigean[3], Liam Sharp[4], Grace Brannigan[4,5], Wayland WL Cheng[1]*

[1]Department of Anesthesiology, Washington University, Saint Louis, United States; [2]Department of Internal Medicine, Mass Spectrometry Resource, Division of Endocrinology, Diabetes, Metabolism, and Lipid Research, Washington University, Saint Louis, United States; [3]Department of Anesthesiology, Weill Cornell Medical College, New York, United States; [4]Center for Computational and Integrative Biology, Rutgers University, Camden, United States; [5]Department of Physics, Rutgers University, Camden, United States

**Abstract** Pentameric ligand-gated ion channels (pLGICs) are essential determinants of synaptic transmission, and are modulated by specific lipids including anionic phospholipids. The exact modulatory effect of anionic phospholipids in pLGICs and the mechanism of this effect are not well understood. Using native mass spectrometry, coarse-grained molecular dynamics simulations and functional assays, we show that the anionic phospholipid, 1-palmitoyl-2-oleoyl phosphatidylglycerol (POPG), preferentially binds to and stabilizes the pLGIC, Erwinia ligand-gated ion channel (ELIC), and decreases ELIC desensitization. Mutations of five arginines located in the interfacial regions of the transmembrane domain (TMD) reduce POPG binding, and a subset of these mutations increase ELIC desensitization. In contrast, a mutation that decreases ELIC desensitization, increases POPG binding. The results support a mechanism by which POPG stabilizes the open state of ELIC relative to the desensitized state by direct binding at specific sites.
DOI: https://doi.org/10.7554/eLife.50766.001

*For correspondence:
wayland.cheng@wustl.edu

**Competing interests:** The authors declare that no competing interests exist.

## Introduction

Pentameric ligand-gated ion channels (pLGICs) are essential determinants of synaptic transmission, and the targets of many allosteric modulators including general anesthetics and anti-epileptics (*Corringer et al., 2012*). These ion channels are embedded in a heterogeneous and dynamic lipid environment (*Allen et al., 2007*), and the presence of specific lipids fine-tunes the function of pLGICs and may play a role in regulating neuronal excitability and drug sensitivity (*Baenziger et al., 2015*; *Rosenhouse-Dantsker et al., 2012*; *Evers et al., 1986*). One nearly ubiquitous example is that of anionic phospholipids, which are known to modulate pLGICs such as the nicotinic acetylcholine receptor (nAchR) (*Criado et al., 1984*), as well as inward rectifying potassium channels (*Cheng et al., 2011*), K(2P) channels (*Chemin et al., 2005*), voltage-gated potassium channels (*Hite et al., 2014*; *Schmidt et al., 2006*), and cyclic nucleotide-gated channels (*Zolles et al., 2006*). In pLGICs, anionic phospholipids have been shown to shift the conformational equilibrium of the channel from an uncoupled or desensitized state to a resting state, in which agonist binding is effectively coupled to channel activation (*daCosta et al., 2009*; *Hamouda et al., 2006*; *daCosta et al., 2004*).

Studies of lipid modulation of ion channel function including modulation of pLGICs have focused on two central questions: 1) what is the exact effect of the lipid on channel function and structure,

and 2) is the effect attributable to direct binding of the lipid at specific sites? *Torpedo* nAchR channel activity measured from flux assays (*Criado et al., 1984*; *Ochoa et al., 1983*; *Fong and McNamee, 1986*) and agonist-induced conformational changes (*Hamouda et al., 2006*; *Baenziger et al., 2000*) depend on anionic phospholipids. However, only a few studies have employed fast solution changes to measure current responses of pLGICs in model membranes (*Velisetty and Chakrapani, 2012*), which is necessary to distinguish the effect of lipids on channel gating, specifically transitions between resting, open, and desensitized states. With regard to lipid binding, early studies using electron paramagnetic resonance (EPR) of spin-labeled lipids or lipid-induced modification of fluorescent probes revealed an immobilized layer of lipids surrounding nAchRs that is enriched for certain phospholipids (*Ellena et al., 1983*; *Mantipragada et al., 2003*) with lipids occupying specific sites (*Dreger et al., 1997*; *Antollini and Barrantes, 1998*). These approaches are, however, an indirect means to examine lipid binding to ion channels. More recently, crystal structures of the pLGIC, Gloeobacter ligand-gated ion channel (GLIC), revealed bound, co-purified phospholipids in a putative open structure, and the absence of one of these phospholipids in a locally closed structure (*Bocquet et al., 2009*; *Prevost et al., 2012*). Similarly, a putative desensitized structure of GLIC with a bound polyunsaturated fatty acid showed loss of the aforementioned phospholipid density that is bound to the open state (*Basak et al., 2017*). Both of these studies suggest that bound phospholipids at specific sites stabilize the open state of the channel, although the identity of these lipids remains unknown. Furthermore, the absence of a lipid density in a crystal structure is not necessarily an indication of lack of binding.

Native mass spectrometry (MS) has proven to be a powerful tool to directly measure binding of endogenous and exogenous lipids to membrane proteins (*Laganowsky et al., 2014*; *Gault et al., 2016*). In addition, coarse-grained molecular dynamics (MD) simulations provide a complementary approach to examine lipid interactions with membrane-embedded pLGICs at time scales that allow equilibration of lipid binding sites (*Sharp et al., 1861*; *Brannigan, 2017*). We sought to determine whether phospholipids bind directly and selectively to a pLGIC by native MS and coarse-grained MD simulations, and whether specific binding interactions modulate channel function by measuring Erwinia ligand-gated ion channel (ELIC) activity in liposomes of defined lipid composition. ELIC, a prototypical pLGIC and biochemically tractable target, is also sensitive to its lipid environment, and was found to be inactive when reconstituted in 1-palmitoyl-2-oleoyl-phosphatidylcholine (POPC) membranes fused to *Xenopus* oocyte membranes, similar to the nAchR (*Carswell et al., 2015a*). After optimizing native MS for ELIC, we demonstrate that phospholipids directly bind to ELIC, with more binding observed for the anionic phospholipid, 1-palmitoyl-2-oleoyl-phosphatidylglycerol (POPG) compared to zwitterionic phospholipids, 1-palmitoyl-2-oleoyl-phosphatidylethanolamine (POPE), and POPC. Consistent with this finding, coarse-grained simulations of ELIC in a lipid bilayer show enrichment of annular POPG compared to POPC or POPE. In addition, POPG selectively stabilizes ELIC against thermal denaturation indicative of a specific binding interaction, and reduces channel desensitization. Mutations of five arginines at the transmembrane domain (TMD) intracellular and extracellular interfaces decrease POPG binding, while a subset of these mutations increase desensitization. The results support the hypothesis that anionic phospholipids stabilize the open state of pLGICs by direct binding to sites in the TMD adjacent to the lipid-facing transmembrane helix 4 (TM4) (*Baenziger et al., 2015*).

## Results

### Selective binding of phospholipids to ELIC

Native MS of ELIC purified in dodecyl maltoside (DDM) was optimized on a Q-Exactive EMR mass spectrometer as previously described (*Gault et al., 2016*). Optimal desolvation of the pentamer required activation energies that resulted in some dissociation into tetramer and monomer (*Figure 1A*). Nevertheless, both the pentamer and tetramer species showed multiple bound small molecules of ~750 Da, likely corresponding to co-purified phospholipids (up to eight and six lipids per multimer were observed for the pentamer and tetramer, respectively) (*Figure 1A*). To determine the identity of these lipids, we performed a lipid extraction from the purified ELIC preparation, and analyzed the sample using tandem MS. This revealed multiple PE and PG phospholipids with different acyl chains that mirror the phospholipids extracted from *Escherichia coli* membranes

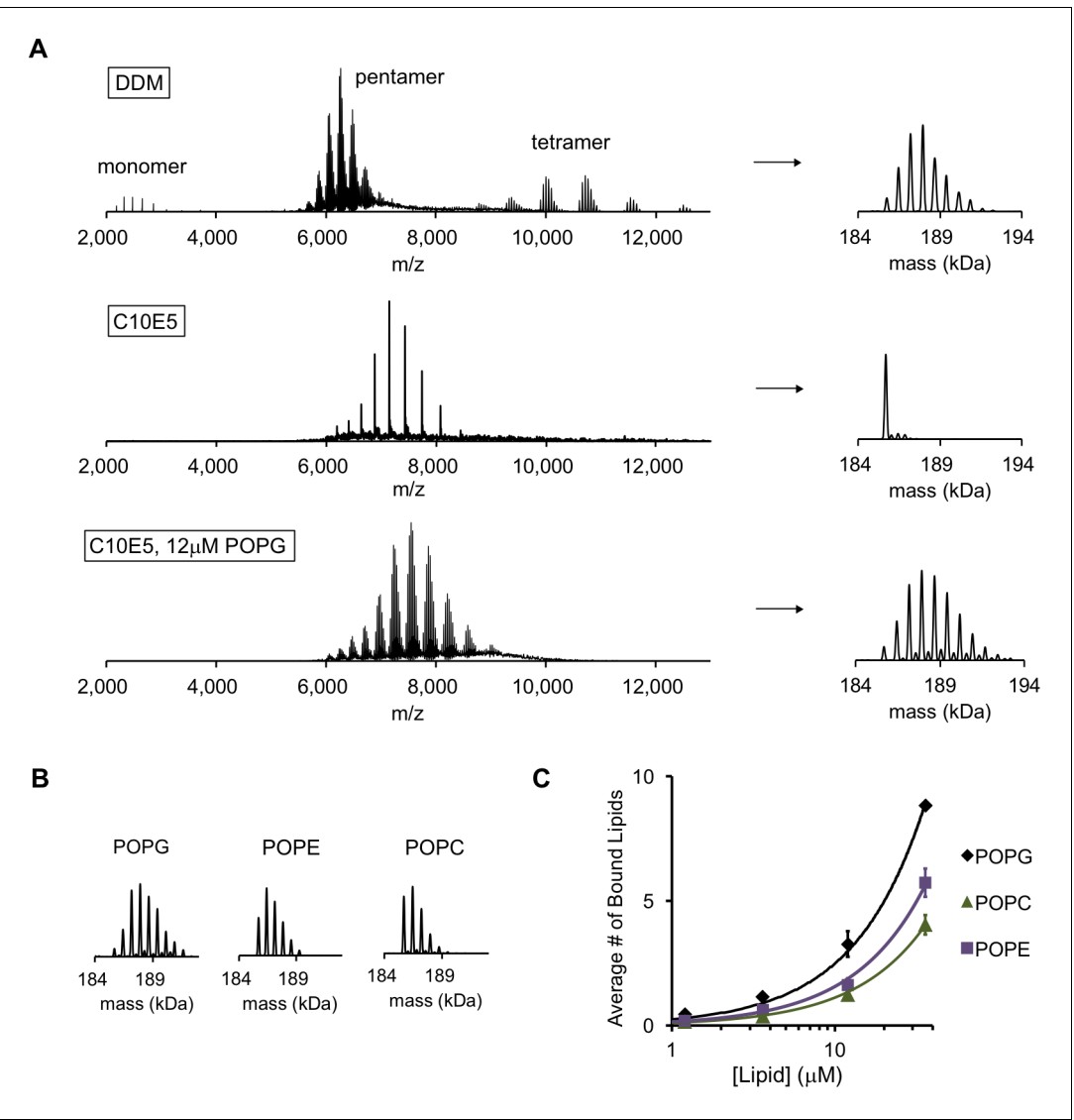

**Figure 1.** POPG binds selectively to ELIC. (**A**) Native MS spectra of ELIC in DDM, C10E5, and C10E5 with 12 μM POPG. *Left:* full spectra; *right:* deconvoluted spectra. (**B**) Deconvoluted spectra of ELIC in 12 μM of the indicated phospholipid. (**C**) Plot of the average number of bound phospholipids per pentamer at varying concentrations of POPG, POPE, and POPC (n = 3–6, ± SD).

DOI: https://doi.org/10.7554/eLife.50766.002

The following figure supplements are available for figure 1:

**Figure supplement 1.** MS1 spectra of lipid extract from purified ELIC in DDM and *E. coli* membranes.

DOI: https://doi.org/10.7554/eLife.50766.003

**Figure supplement 2.** Representative deconvoluted spectra of 1 μM ELIC in C10E5 with increasing concentration of POPG.

DOI: https://doi.org/10.7554/eLife.50766.004

**Figure supplement 3.** Comparison of lipid binding at different charge states.

DOI: https://doi.org/10.7554/eLife.50766.005

**Figure supplement 4.** Lipid binding data fit to binomial distributions.

DOI: https://doi.org/10.7554/eLife.50766.006

(*Supplementary file 1*). Quantification of the MS intensities for PG (phosphatidylglycerol) relative to PE (phosphatidylethanomamine) species yielded a higher relative abundance of PG co-purified with ELIC compared to *E. coli* membranes, suggesting that ELIC preferentially binds PG in its native environment (*Figure 1—figure supplement 1*, *Supplementary file 1*).

To examine direct binding of exogenous phospholipids to ELIC, we performed a detergent screen to delipidate ELIC focusing on detergents that are also superior for native MS measurements (*Reading et al., 2015*). The polyethylene glycol-based detergent, C10E5, proved best for this application, yielding a stable, delipidated pentamer by native MS with lower charge states and no dissociation of the pentamer (*Figure 1A*). This observation is consistent with previous reports in other membrane proteins (*Reading et al., 2015*; *Liu et al., 2019*). Addition of varying concentrations of the anionic phospholipid, POPG, to 1 µM ELIC showed concentration-dependent binding (*Figure 1—figure supplement 2*). We quantified this binding by calculating the average number of bound phospholipids at each concentration. For example, at 12 µM POPG, native MS spectra revealed up to nine bound POPG per pentamer or an average of 2.9 POPG per pentamer (*Figure 1B and C*). The average number of bound POPG was equivalent for most charge states, and decreased modestly at charge states > 26+ likely because of electrostatic repulsion within the ELIC-POPG complexes (*Figure 1—figure supplement 3*); therefore, deconvolution was performed for charge states ≤ 26+. Less binding was observed for the neutral phospholipids, POPE and POPC (*Figure 1B and C*), indicating that the anionic phospholipid, POPG, either binds with higher affinity or at more sites.

To further examine phospholipid interactions with ELIC using a molecular model, we performed coarse-grained MD simulations on binary POPG/POPC and POPG/POPE model membranes containing a single ELIC pentamer (*Figure 2A*). Unlike fully atomistic simulations, coarse-grained simulations permit significant diffusion of lipids over simulation time scales. The boundary lipid composition can thus equilibrate over the simulation time, even if it varies significantly from the bulk membrane composition. The POPG fraction was varied between 0 and 70%. Enrichment or depletion of POPG among boundary lipids for each concentration was quantified using the boundary lipid metric B (*Equation 8*, see Materials and methods). For a given lipid species, B > 0 reflects enrichment, B < 0 reflects depletion, and B = 0 reflects random mixing. For POPG, B>0 for all compositions tested (*Figure 2B*). This result indicates that if POPG is present in the membrane, it is enriched among

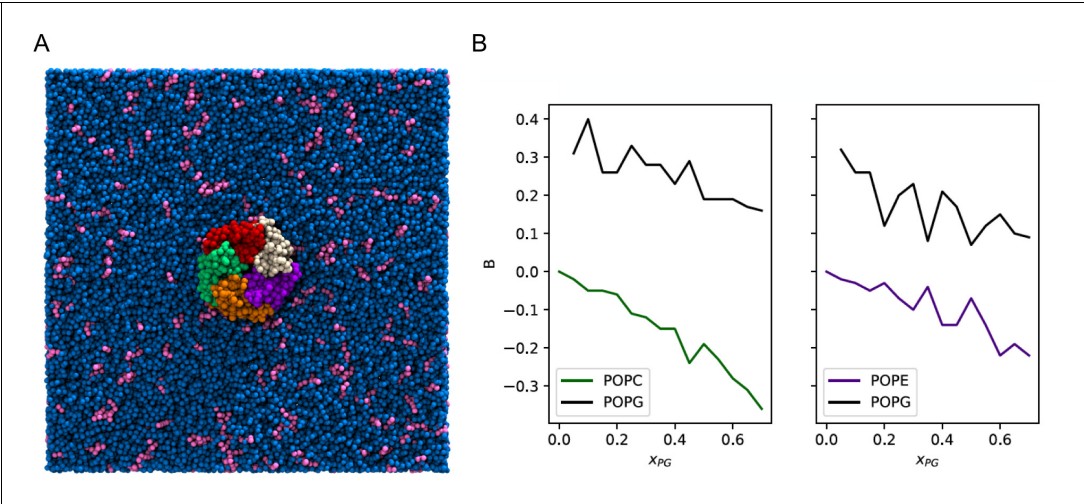

**Figure 2.** Enrichment of POPG among ELIC boundary phospholipids from coarse-grained simulations. (A) Image of the simulation model of ELIC embedded in a membrane consisting of 10% POPG (pink) and 90% POPC (blue). The view is from the extracellular side of ELIC perpendicular to the membrane. (B) The boundary enrichment metric, B, is shown for phospholipid species in POPC/POPG membranes (left) or POPE/POPG membranes (right) over a range of POPG mole fractions ($x_{PG}$)). B is defined in *Equation 8* (see Materials and methods) and reflects the fractional difference between the amount of a lipid species found in the boundary and the bulk membrane: B > 0 indicates enrichment, B < 0 indicates depletion, and B = 0 indicates no difference in mole fraction between the bulk and the boundary.
DOI: https://doi.org/10.7554/eLife.50766.007

boundary lipids. This enrichment is strongest for lower amounts of POPG (i.e. lower $x_{PG}$), consistent with specific binding of POPG to ELIC.

The average number of boundary phospholipids was $31.6 \pm 2.5$ ($\pm$ SD) across all compositions, and the total did not vary systematically with membrane composition. Therefore, we assumed that the stoichiometries of binding for these phospholipids to ELIC are similar, and fit the native MS binding data for each phospholipid to a binomial distribution binding model with 32 binding sites of equivalent affinity (see Materials and methods). While this is an oversimplification of phospholipid binding to ELIC in a membrane, it provides a reasonable approximation to the MS data, and reveals that POPG binds to ELIC with ~1.9× and 2.8× higher affinity than POPE and POPC, respectively (*Figure 1—figure supplement 4*). Overall, we conclude that POPG binds to ELIC with higher affinity than POPE or POPC, resulting in POPG enrichment of annular phospholipids as seen in the coarse-grained MD simulations.

## Selective effect of POPG on ELIC stability and function

To determine the effect of POPG binding on ELIC, we first tested the stability of purified, delipidated ELIC in C10E5 against thermal denaturation in the absence and presence of POPG (*Nji et al., 2018*). ELIC was heated to a temperature that resulted in 85% decrease in the amplitude of the pentamer peak as assessed by size exclusion chromatography (32°C for 15 min). POPG significantly increased the thermal stability of 1 μM ELIC with an $EC_{50}$ (concentration of POPG for 50% effect) of 52 μM (*Figure 3A*). The thermal stabilizing effect of a phospholipid was defined as the ratio of the pentamer peak height after heating with lipid versus no lipid. In contrast, POPE and POPC had no effect on ELIC stability (*Figure 3A*), indicating that POPG binding selectively stabilizes the structure of ELIC. Having performed our POPG binding experiment and thermal stability assay under the same conditions, it is possible to relate the average number of bound POPG to its stabilizing effect. The highest concentration for which the average number of bound POPG could be determined because of overlapping of charge states from lipid-bound species was 36 μM (*Figure 1A*, *Figure 1—figure supplement 2*). Although POPG binding does not approach saturation at this concentration, extrapolation of POPG binding and relating this extrapolation to the thermal stabilizing effect provides an approximation of the number of bound POPG needed to stabilize ELIC against thermal denaturation. *Figure 3—figure supplement 1* shows a relationship between the number of bound POPG and the stabilizing effect, which was derived by equating the POPG concentration from the functions of POPG binding (*Figure 1C*) and thermal stability data (*Figure 3A*). The relationship estimates that 32 POPG (number of annular lipids in ELIC from MD simulations) yield ~82% of the thermal stabilizing effect (*Figure 3—figure supplement 1*).

Next, we assessed the effect of POPG on ELIC function by reconstituting the channel in giant liposomes. Optimal formation of giant liposomes was achieved using a 2:1:1 ratio of POPC:POPE:POPG (i.e. 25 mole% POPG). In this lipid membrane composition, robust ELIC currents were elicited with excised patch-clamp recordings using the agonist, cysteamine, with a peak dose response $EC_{50}$ of 5.1 mM (*Figure 3B*, *Table 1*, *Figure 3—figure supplement 2A*). Patch-clamp recordings were performed with 0.5 mM $BaCl_2$ in the pipette and bath, which is predicted to result in an increase in the $EC_{50}$ of cysteamine response (*Zimmermann et al., 2012*). Near saturating currents were achieved at 30 mM cysteamine at which ELIC activated and desensitized with time constants of 134 ms and 1.9 s, respectively (*Figure 3C and D*, *Table 1*, *Figure 3—figure supplement 2B*). These values are comparable to previous reports of outside-out patch-clamp recordings in HEK cells or oocytes (*Laha et al., 2013*; *Gonzalez-Gutierrez et al., 2012*). ELIC desensitization showed complex kinetics where the majority of recordings were best fit with a double exponential and some by a single exponential. To combine data from all traces, weighted average time constants from double exponential fits were averaged with time constants from single exponential fits. The extent of desensitization was examined by measuring currents after 20 s of cysteamine application. To examine the effect of POPG on ELIC gating, excised patch-clamp recordings were performed in liposomes containing 12%, 25%, and 40% POPG. Increasing the mole% of POPG had no significant effect on cysteamine $EC_{50}$ values or activation kinetics (*Figure 3B*, *Table 1*, *Figure 3—figure supplement 2*), but reduced the rate and extent of desensitization (*Figure 3C and D*, *Table 1*).

To examine ELIC activity in the absence of POPG, a fluorescence-based stopped-flow flux assay was performed (*Posson et al., 2018*). ELIC was reconstituted into either POPC-alone or 2:1:1 POPC:POPE:POPG liposomes encapsulating the fluorophore ANTS (8-aminonaphthalene-1,3,6-trisulfonic

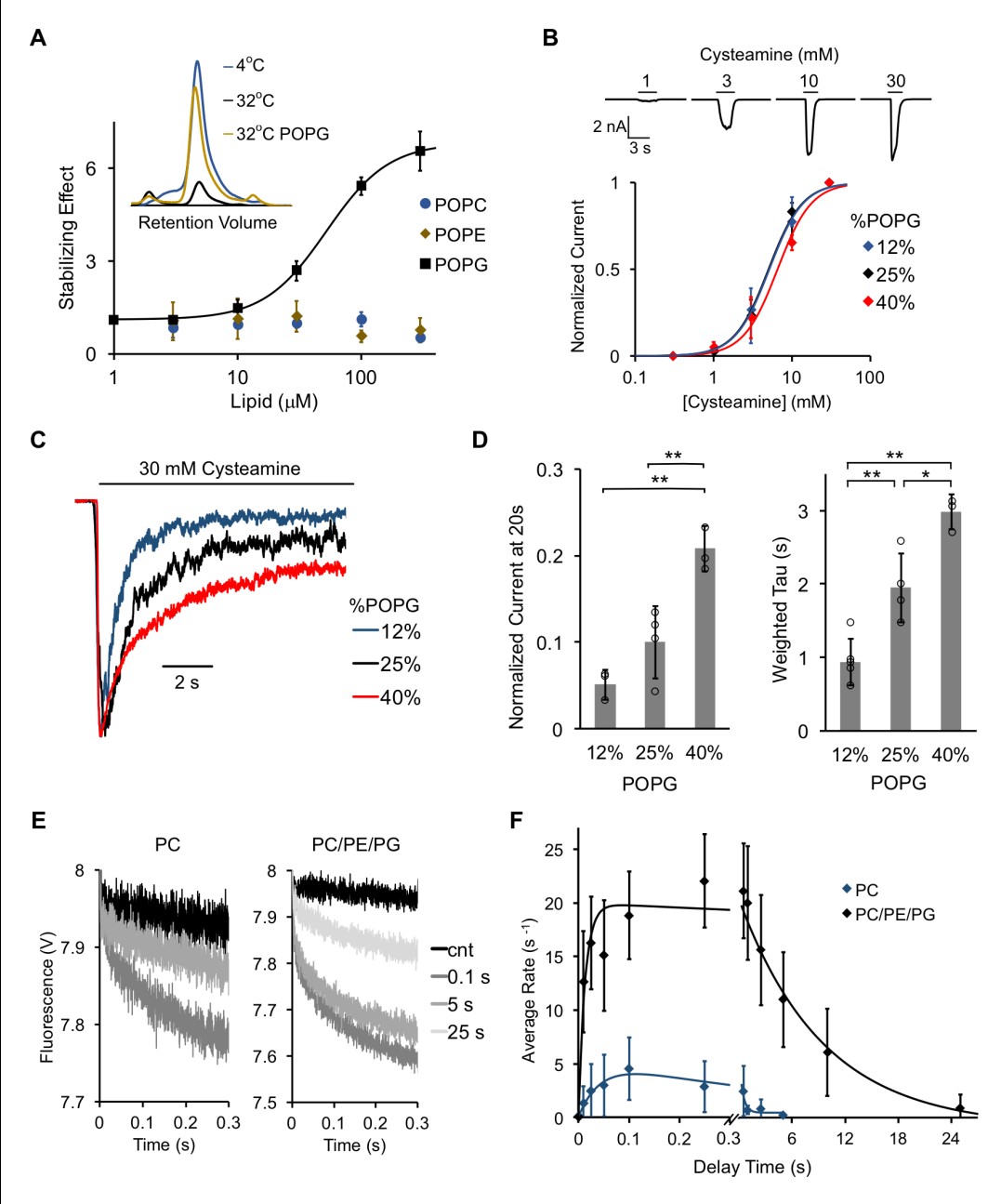

**Figure 3.** POPG selectively thermally stabilizes ELIC and decreases ELIC desensitization. (A) Plot of stabilizing effect (defined as the ELIC pentamer peak height with phospholipid relative to control after heating) versus phospholipid concentration (n = 3, ± SD; EC$_{50}$ = 52 μM, Hill n = 1.7). Inset shows size exclusion chromatography (SEC) profile in absorbance units of the ELIC pentamer treated at 4°C, 32°C, and 32°C with 100 μM POPG. (B) *Top:* Representative ELIC current responses to 30 mM cysteamine in 25 mole% POPG liposomes. *Bottom:* Normalized plots of peak current responses of ELIC to cysteamine in giant liposomes with varying mole% POPG (n = 3–5, ± SD). Data are fit to Hill equation with n = 2. (C) Representative ELIC currents in response to 30 mM cysteamine in liposomes with varying mole% POPG. (D) *Left:* ELIC currents 20 s after application of 30 mM cysteamine normalized to peak response at varying mole% POPG (n = 4–5, ± SD, **p<0.01). *Right:* Weighted tau (time constants) of ELIC desensitization at varying mole% POPG (n = 3–5, ± SD, **p<0.01, *p<0.05). (E) Representative fluorescence-quench time courses from sequential mixing stopped-flow experiments of ELIC in POPC liposomes or 2:1:1 POPC:POPE:POPG liposomes. Proteoliposomes were mixed with no cysteamine (cnt) or 5 mM cysteamine with a 0.1, 5, and 25 s delay prior to mixing with Tl$^+$. (F) Rate constants extracted from quench kinetics as shown in

*Figure 3 continued on next page*

*Figure 3 continued*

(E) as a function of the incubation time with cysteamine. Data are fit with a double exponential yielding activation and desensitization time constants (n = 3, ± SD).

DOI: https://doi.org/10.7554/eLife.50766.008

The following figure supplements are available for figure 3:

**Figure supplement 1.** Relationship of the thermal stabilizing effect of POPG vs the average number of bound POPG derived from equating concentration of POPG from the sigmoid functions used to fit the POPG binding (*Figure 1B*) and thermal stability data (*Figure 3A*).

DOI: https://doi.org/10.7554/eLife.50766.009

**Figure supplement 2.** Channel properties of WT ELIC responses to cysteamine by patch-clamping.

DOI: https://doi.org/10.7554/eLife.50766.010

acid). In a first mixing step, liposomes were incubated with 5 mM cysteamine to activate the channel for different amounts of time (10 ms to 25 s), after which a second mixing step was performed with $Tl^+$ containing buffer. $Tl^+$ can permeate through activated channels into the liposomes where it quenches ANTS fluorescence. The quenching kinetics are a measure of the channel activity upon cysteamine exposure for defined incubation times (*Figure 3E*). In POPC liposomes, ELIC showed less cysteamine-elicited ion flux compared to ELIC in POPC:POPE:POPG liposomes (*Figure 3E and F*, *Table 2*), as estimated from the overall rate of $Tl^+$ flux. The rate of activation was modestly faster in POPC:POPE:POPG liposomes compared to POPC (*Figure 3F*, *Table 2*). More strikingly, the rate of desensitization was more than 20-fold faster in POPC liposomes, leading to a decrease in the lifetime of the open state (*Figure 3E and F*, *Table 2*).

In summary, POPG selectively increases the thermal stability of ELIC, and modulates channel activity by stabilizing the open relative to the desensitized state. We hypothesize that POPG decreases receptor desensitization by direct binding at specific sites.

## Five interfacial arginines contribute to POPG binding

In other ion channels, guanidine groups from interfacial arginine side chains are thought to mediate binding of anionic phospholipids by charge interactions with the phospholipid headgroup (*Hite et al., 2014*; *Lee et al., 2013*). To test the hypothesis that this mechanism is present in a pLGIC, we mutated all five arginines in the inner and outer interfacial regions of the ELIC TMD to glutamine (*Figure 4A*). Phospholipid binding was then assessed by delipidating each mutant in C10E5, and measuring binding of POPG by native MS. While R123Q, R286Q, and R299Q could be stably delipidated, R117Q and R301Q aggregated (*Figure 4B*). However, we found that double mutants with R299Q (i.e. R117Q/R299Q and R301Q/R299Q) could be stably delipidated. Thus, double mutants of all arginine mutants in combination with R299Q were expressed and delipidated (*Figure 4B*). In the presence of 12 µM POPG, the single mutants showed moderate decreases (13–18%) in the average number of bound POPG compared to WT (*Figure 4B*). This decrease was not statistically significant. However, all double mutants significantly decreased the average number of

**Table 1.** ELIC WT channel properties by patch-clamping giant liposomes composed of varying mole % POPG.

The rate and extent of desensitization are reported as weighted time constants ($\tau$), and the current after 20 s of 30 mM cysteamine application normalized to peak response. Also shown are activation time constants ($\tau$) in response to 30 mM cysteamine and $EC_{50}$s for cysteamine activation (n = 3–5, ± SD).

| POPG | Desensitization | | Activation | Dose response |
|------|------|------|------|------|
| | Weighted $\tau$ (s) | Norm current at 20 s | $\tau$ (ms) | $EC_{50}$ (mM cysteamine) |
| 12% | 0.93 ± 0.32 | 0.05 ± 0.02 | 112 ± 29 | 5.3 ± 1.0 |
| 25% | 1.95 ± 0.47 | 0.10 ± 0.04 | 134 ± 50 | 5.1 ± 1.2 |
| 40% | 2.98 ± 0.24 | 0.21 ± 0.03 | 133 ± 66 | 6.5 ± 1.3 |

DOI: https://doi.org/10.7554/eLife.50766.011

**Table 2.** ELIC WT activation and desensitization time constants (τ) derived from a double exponential fit to the entire time course of flux in **Figure 3F** (n = 3, ± SD). The fast component describes activation and the slow component desensitization.

| | Activation τ (ms) | Desensitization τ (s) |
|---|---|---|
| POPC | 39 ± 17 | 0.42 ± 0.05 ** |
| POPC:POPE:POPG (2:1:1) | 13 ± 3 | 9.2 ± 3.2 ** |

** indicates a significant difference between desensitization time constants in POPC compared to POPC:POPE:POPG (p<0.01).

DOI: https://doi.org/10.7554/eLife.50766.012

bound POPG relative to WT by 38–41% and relative to R299Q by 27–32% (**Figure 4B**). These results indicate that each interfacial arginine contributes approximately equally to POPG binding in ELIC. It is likely that significant decreases in binding could only be appreciated in the double mutants because of the variability in the data.

We further examined these sites of interaction using our coarse-grained MD simulations. To identify whether boundary POPG were localized around specific helices or residues, two-dimensional densities of the negatively charged headgroup bead were calculated. The distributions are separated by leaflet where each leaflet contained 10% POPG. As shown in **Figure 5A**, POPG was more likely to interact with ELIC in the inner leaflet than the outer leaflet, consistent with three out of five interfacial arginine residues being located on the intracellular interface of the ELIC TMD. These three arginines are located on TM3 (R286) and TM4 (R299 and R301). Contacts between POPG and all three of these residues are also visible in individual frames of the simulation (**Figure 5B**). Moreover, POPG is more likely to be contacting the interfacial residues in TM4 (such as R299 and R301) than accessible interfacial residues in any other helix (**Figure 5A**). The remaining two arginine residues

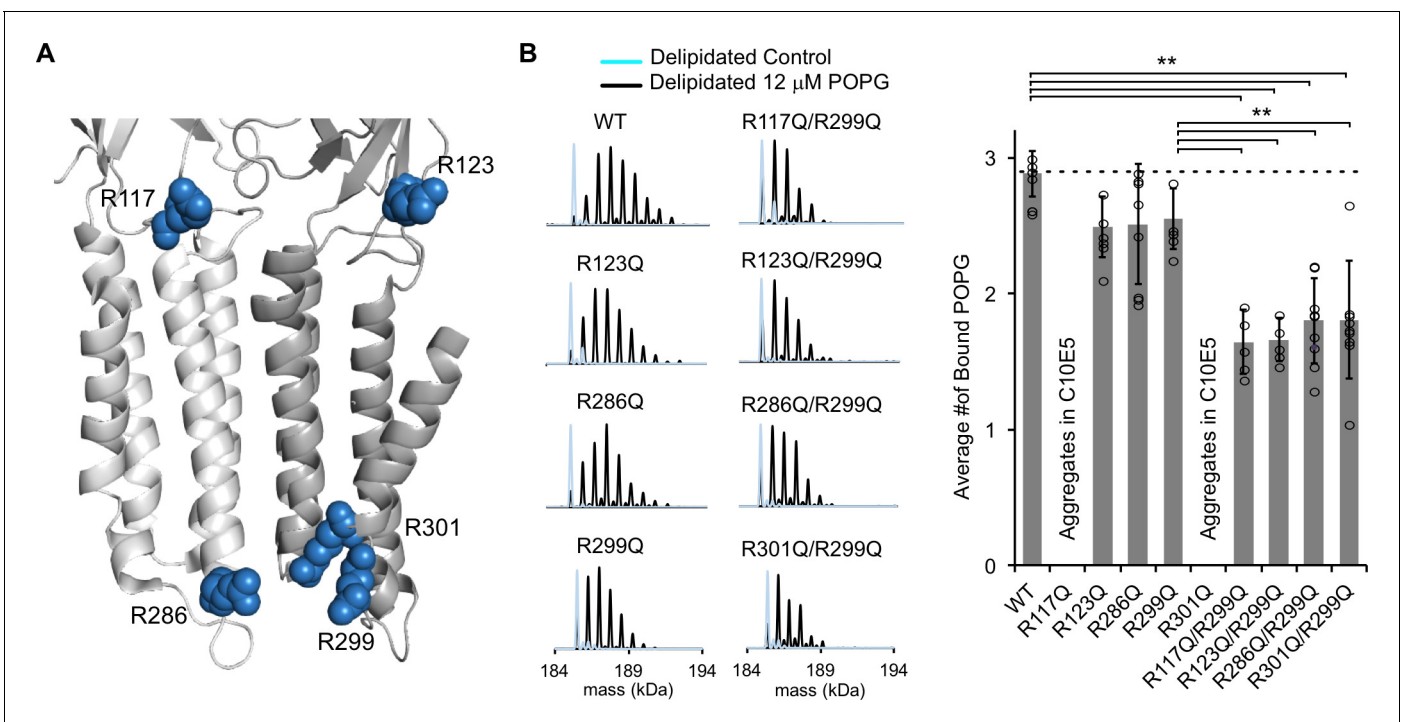

**Figure 4.** Mutations of five interfacial arginines reduce POPG binding. (**A**) Structure of ELIC showing two adjacent subunits and five arginine side chains that were mutated to glutamine. (**B**) *Left:* Representative deconvoluted spectra of ELIC WT and indicated mutants. Blue indicates spectra of delipidated ELIC in C10E5. Black indicates spectra of delipidated ELIC in C10E5 with 12 μM POPG. *Right:* Plot of average number of bound POPG for ELIC WT and mutants, delipidated in C10E5, with 12 μM POPG (n = 5–8, ± SD, **p<0.01).

DOI: https://doi.org/10.7554/eLife.50766.013

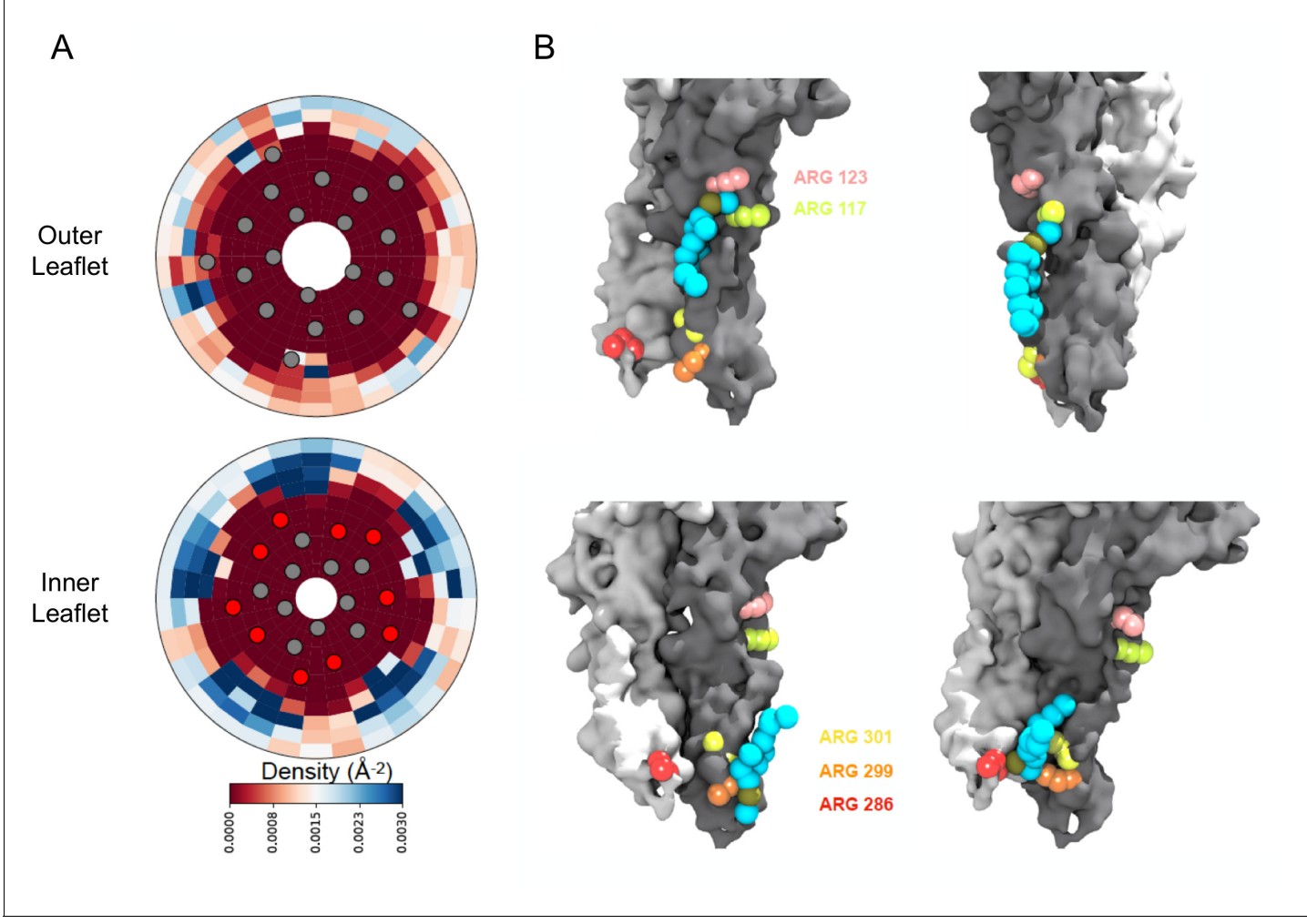

**Figure 5.** Density calculations of lipids in binary membranes and visualization of direct POPG-ELIC interactions at 10% POPG. (**A**) Distribution of POPG density in a POPG-POPC membrane, within 40 Å from the ELIC pore over the last half of a 15 μs simulation, for both the outer leaflet (*top*) and the inner leaflet (*bottom*). Density is colored according to the color bar, where red and blue represent low and high POPG density, respectively. Circles represent the ELIC transmembrane backbone center of mass, with the helices containing the interfacial arginines colored in red. (**B**) Representative frames after ~9 μs of simulation, showing multiple POPG binding modes associated with high density areas in (**A**). Two adjacent subunits of ELIC are shown in gray and white, while arginine side chains of interest are colored in peach, lime-yellow, orange, yellow, and red. POPG phosphate is colored in tan with the rest of the lipid in cyan.

DOI: https://doi.org/10.7554/eLife.50766.014

are located at the TMD-extracellular domaininterface (R117 and R123). POPG density in the outer leaflet localized to these residues at intrasubunit sites between TM4 and TM1 or TM4 and TM3 (*Figure 5A*), and contacts between these residues and POPG headgroups in the outer leaflet were also observed in snapshots from the MD simulations (*Figure 5B*). In summary, the native MS data and coarse-grained MD simulations demonstrate that five interfacial arginines contribute to specific POPG binding sites in the inner and outer leaflets adjacent to TM4.

## Specific interfacial arginines mediate POPG effect

Having established that ELIC selectively binds POPG over neutral phospholipids, and that binding is mediated by five interfacial arginines, we examined the role of these binding sites on ELIC desensitization. We reconstituted each single mutant into giant liposomes composed of a 2:1:1 ratio of POPC:POPE:POPG (25% POPG) to test channel function by excised patch-clamping. We hypothesized that because increasing mole% POPG decreases ELIC desensitization, certain arginine mutants,

which disrupt POPG binding, may increase ELIC desensitization. Indeed, all five single arginine mutants showed variable increases in the rate or extent of desensitization; however, these differences were generally small and statistically insignificant except for R301Q (*Figure 6*, *Table 3*). We also tested the double mutants, which showed significant decreases in POPG binding. Three double mutants (R117Q/R299Q, R123Q/R299Q, R301Q/R299Q) showed a significant increase in the extent of desensitization, while two (R117Q/R299Q, R301Q/R299Q) also showed a significant increase in the rate of desensitization (*Figure 6*, *Table 3*). The effects observed in the double mutants approximate the sum of effects observed in the single mutants. Only R286Q/R299Q did not affect the extent or rate of desensitization (*Figure 6*, *Table 3*). The EC$_{50}$ of cysteamine response and activation kinetics were also measured for all mutants; only R117Q and R117Q/R299Q showed significantly lower EC$_{50}$ and activation tau values compared to WT (*Table 3*, *Figure 6—figure supplement 1*). Overall, these data indicate that four of five interfacial arginine residues that reduce POPG binding (i.e. R117, R123, R299, R301) also increase the rate and/or extent of ELIC desensitization.

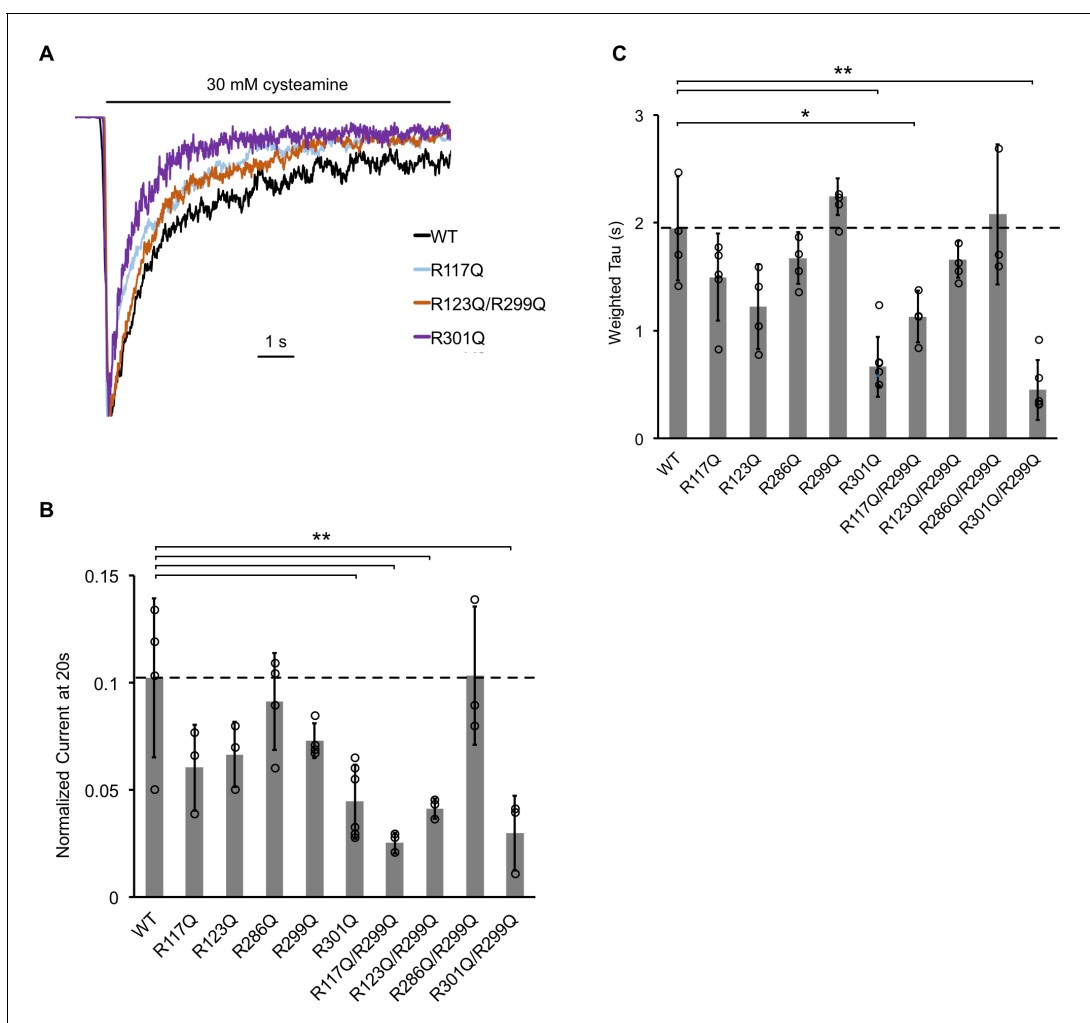

**Figure 6.** The effect of ELIC mutants on desensitization. (**A**) Normalized ELIC WT and mutant current responses to 30 mM cysteamine in 25 mole% POPG liposomes. (**B**) Graph of ELIC WT and mutant currents 20 s after application of 30 mM cysteamine normalized to peak response in 25 mole% POPG liposomes (n = 3–7, ± SD, **p<0.01, *p<0.05). (**C**) Same as (**B**) for weighted tau (time constants) of desensitization.
DOI: https://doi.org/10.7554/eLife.50766.015

The following figure supplement is available for figure 6:

**Figure supplement 1.** Channel properties of ELIC WT and mutant responses to cysteamine b giant liposomes of 25 mole% POPG.
DOI: https://doi.org/10.7554/eLife.50766.016

**Table 3.** ELIC WT and mutant channel properties by patch-clamping giant liposomes composed of 25 mole% POPG (n = 3–7, ± SD). Shown are weighted time constants (τ) for desensitization and currents 20 s after application of 30 mM cysteamine normalized to peak response. Also shown are activation time constants and EC$_{50}$ of cysteamine response. Light gray indicates mutant values which are significantly different from WT (blue) (\*\*p<0.01, \*p<0.05).

| | Desensitization | | Activation | Cysteamine response |
|---|---|---|---|---|
| | Weighted τ (s) | Norm current at 20 s | τ (ms) | EC$_{50}$ (mM cysteamine) |
| WT | 1.95 ± 0.48 | 0.100 ± 0.041 | 134 ± 50 | 5.1 ± 1.2 |
| R117Q | 1.49 ± 0.40 | 0.060 ± 0.020 | 54 ± 40 * | 3.3 ± 0.9 * |
| R123Q | 1.20 ± 0.39 | 0.067 ± 0.015 | 112 ± 65 | 3.7 ± 0.9 |
| R286Q | 1.67 ± 0.24 | 0.091 ± 0.023 | 106 ± 34 | 4.3 ± 0.4 |
| R299Q | 2.2 ± 0.17 | 0.074 ± 0.008 | 71 ± 27 | 4.7 ± 0.3 |
| R301Q | 0.66 ± 0.28 ** | 0.045 ± 0.017 ** | 72 ± 15 | 5.2 ± 0.8 |
| R117Q/R299Q | 1.13 ± 0.24 * | 0.025 ± 0.004 ** | 54 ± 26 * | 2.4 ± 0.7 ** |
| R123Q/R299Q | 1.66 ± 0.17 | 0.041 ± 0.005 ** | 74 ± 35 | 3.7 ± 0.6 |
| R286Q/R299Q | 2.08 ± 0.65 | 0.103 ± 0.032 | 92 ± 27 | 4.7 ± 1.5 |
| R301Q/R299Q | 0.45 ± 0.28 ** | 0.030 ± 0.017 ** | 71 ± 21 | 3.9 ± 0.9 |
| L238A | 5.90 ± 1.37 ** | 0.374 ± 0.045 ** | 113 ± 51 | 4.9 ± 1.1 |

DOI: https://doi.org/10.7554/eLife.50766.017

## A mutation that reduces desensitization enhances POPG binding

If mutations that disrupt POPG binding increase receptor desensitization, then a mutation that decreases desensitization may enhance POPG binding. While characterizing other ELIC mutations, we discovered that mutation of the 7' TM2 leucine residue (L238A) markedly decreased the rate and extent of desensitization in response to 30 mM cysteamine (*Figure 7A*), without significantly altering the EC$_{50}$ of cysteamine response or activation time constant (*Figure 6—figure supplement 1*, *Table 3*). To examine POPG binding, L238A was then delipidated in C10E5 for native MS. Interestingly, L238A significantly increased POPG binding compared to WT at 12 mM POPG (~1.7x increase in average number of POPG bound; *Figure 7B*).

## Discussion

Recent structural and computational evidence suggests that lipids bind to pLGICs at specific sites within the TMD (*Prevost et al., 2012*; *Basak et al., 2017*; *Althoff et al., 2014*; *Laverty et al., 2019*; *Carswell et al., 2015b*). However, there is a scarcity of evidence showing that changes in direct lipid binding are correlated with functional effects (*Basak et al., 2017*). We show that the anionic phospholipid, POPG, selectively binds to ELIC using native MS, thermally stabilizes the channel, and decreases receptor desensitization. Overall, these data support the idea that lipid binding directly affects receptor stability and function. Further, mutations of arginine residues that reduce POPG binding also increase ELIC desensitization to varying degrees. While it is possible that these arginine mutations increase desensitization through a mechanism other than their effect on POPG binding, the correlation between binding and desensitization, and the finding that L238A, which reduces desensitization, increases POPG binding affinity supports this conclusion. This mutation, which is remote from the lipid interface (*Figure 8*), appears to allosterically increase the affinity or number of binding sites for POPG in ELIC. Lipids may modulate ion channel activity through indirect effects on the physical properties of the membrane or through direct binding interactions (*Cordero-Morales and Vásquez, 2018*; *daCosta et al., 2013*). The lipid binding data presented in this study using native MS provide evidence that direct binding of anionic phospholipids allosterically stabilizes the open state of a pLGIC relative to the desensitized state.

Membrane proteins including pLGICs are thought to determine their lipid microenvironment by specific binding interactions (*Sharp et al., 1861*; *Patrick et al., 2018*). Our native MS measurements provide unique insights into phospholipid interaction with a pLGIC. First, we find that more POPG binds to ELIC compared to POPE or POPC at equivalent concentrations, suggesting that POPG

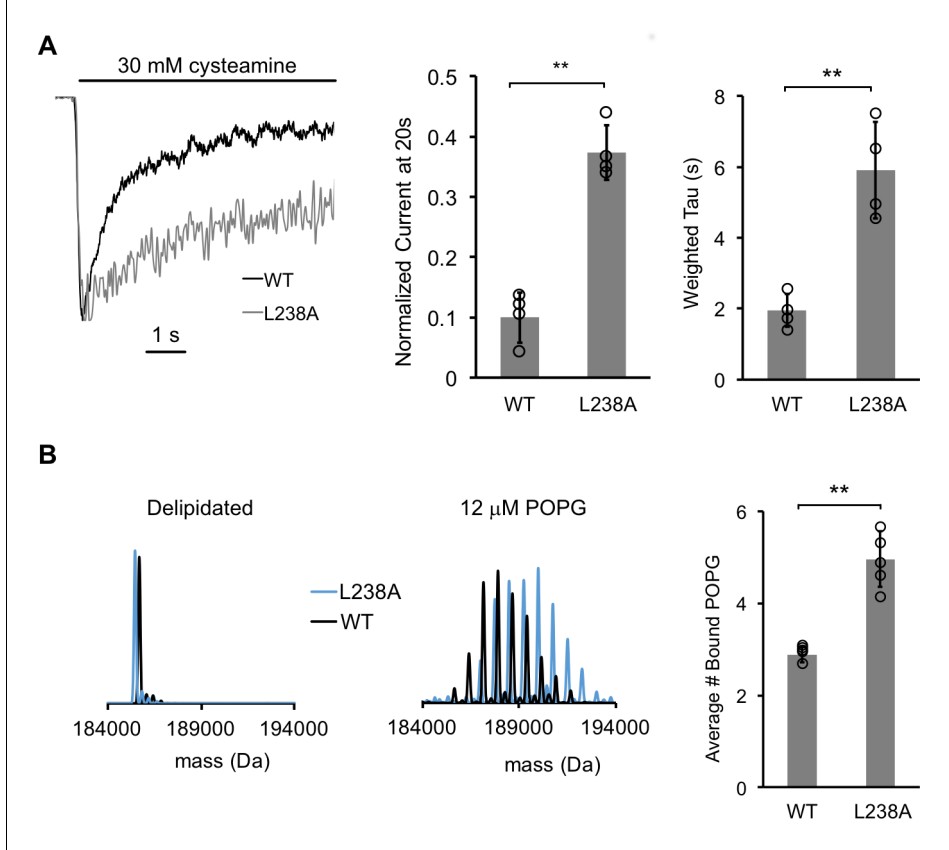

**Figure 7.** A mutantion, which decreases desensitization, increases POPG binding. (**A**) *Left:* Normalized ELIC WT and L238A current responses to 30 mM cysteamine in 25% POPG liposomes. *Middle:* ELIC WT and L238A currents 20 s after application of 30 mM cysteamine normalized to peak response at 25 mole% POPG (n = 4–5, ± SD, **p<0.01). *Right:* Weighted tau (time constants) of ELIC WT and L238A desensitization time courses at 25 mole% POPG (n = 4–5, ± SD, **p<0.01). (**B**) *Left:* Representative deconvoluted spectra of ELIC WT (black) and L238A (blue) showing ELIC delipidated in C10E5 without and with 12 μM POPG. *Right:* Graph of average number of bound POPG for ELIC WT and L238A, delipidated in C10E5, with 12 μM POPG (n = 4–5, ± SD, **p<0.01).
DOI: https://doi.org/10.7554/eLife.50766.018

binds to ELIC with higher affinity. This is supported by enrichment of POPG compared to POPE in phospholipids that are co-purified with ELIC, and coarse-grained simulations which show enrichment of POPG among the boundary phospholipids of ELIC. Second, native MS also allows determination of the stoichiometry and sites of lipid binding (*Liu et al., 2019*; *Habeck et al., 2017*). By relating binding stoichiometry and thermal stability, the data estimate that 32 POPG lipids, which is the average number of annular lipids in ELIC from MD simulations, result in > 80% of the stabilizing effect against thermal denaturation (*Figure 3—figure supplement 1*). This suggests that maximal thermal stability is achieved when the entire ELIC TMD is surrounded by POPG. Although five interfacial arginine residues were identified to contribute to POPG binding in ELIC (25 arginines total), it is conceivable that each arginine side chain may interact with more than one phospholipid headgroup or that other sites exist.

To quantify the effect of the ELIC double mutants on specific POPG binding sites, we also fit the native MS binding data for the double mutants to a binomial binding model using the dissociation constant for POPG binding to WT and varying the number of available sites. POPG binding to the double mutants was best fit with a reduction in the number of available sites from 32 in WT to 18–21 in the mutants (*Figure 1—figure supplement 4*). Given the ~35–45% decrease in bound phospholipid with each double mutant (mutation of two out of five arginines), we infer that phospholipid binding at these residues constitute the highest affinity sites. Alternatively, the arginine mutations could have indirect allosteric effects on other POPG binding sites. Nevertheless, the native MS data

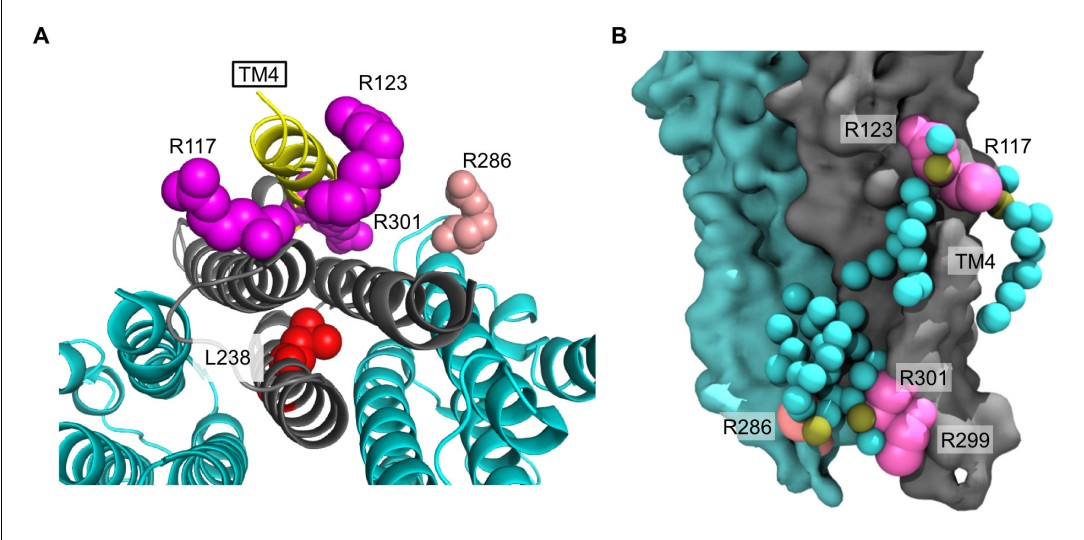

**Figure 8.** Arginines involved in POPG binding and ELIC desensitization. (**A**) Top view of ELIC highlighting TM4 (yellow) and showing the side chains of R117, R123, and R301 (magenta) adjacent to TM4, which increase ELIC desensitization, and R286 at the subunit interface (salmon), which has no effect on desensitization. The L238 side chain is shown in red. (**B**) Image from coarse-grained simulations with 50% POPG showing two adjacent ELIC subunits and the mutated arginine side chains (R117, R123, R299, and R301 in magenta; R286 in salmon). Also shown are all POPG lipids making contacts with the TMD in this snapshot.
DOI: https://doi.org/10.7554/eLife.50766.019

for POPG binding in the arginine mutants are consistent with the localization of POPG in the coarse-grained simulations, which show discrete enrichment of POPG adjacent to these residues. Disruption of binding sites by mutation of these arginines may not prevent the occupancy of lipids at these sites per se, but may alter the lipid binding modes or occupancy times at these sites.

We observed POPG binding to ELIC by native MS using positive ion mode. It was previously shown that phospholipid binding to certain membrane proteins varies significantly when using positive versus negative ion mode (i.e. a positively versus negatively charged protein), and that this difference accounts for pH-dependent changes in the function of the outer membrane protein, OmpF (*Liko et al., 2018*). While phospholipid binding to ELIC was not examined using negative ion mode in this study, our mutagenesis results suggest that POPG binding is dependent on protein charge. Therefore, titration of the protonation state of certain residues such as the arginines examined in this study may underlie pH-dependent changes in ELIC gating (*Gonzalez-Gutierrez et al., 2012*).

Previous studies examining the effects of lipids on pLGIC function found that nAchR and ELIC are inactive in POPC-only membranes (*Criado et al., 1984*; *Ochoa et al., 1983*; *Carswell et al., 2015a*), and it was proposed that this results from uncoupling of agonist binding to channel activation (*daCosta and Baenziger, 2009*). To examine ELIC channel activity in liposomes lacking anionic phospholipid, we utilized a stopped-flow flux assay, and demonstrated cysteamine-elicited flux by ELIC in POPC-only liposomes. The high sensitivity of this assay may be the reason ELIC activity could be detected, contrary to a prior study in which ELIC in POPC liposomes were injected into *Xenopus* oocytes (*Carswell et al., 2015a*). However, the ELIC activity was significantly decreased compared to POPC:POPE:POPG (2:1:1) liposomes. The low protein concentration used in this assay does not allow us to assess the reconstitution efficiencies. Thus, the overall lower flux rates and smaller amplitudes in POPC could stem from lower protein reconstitution. However, the faster desensitization kinetics in POPC liposomes can be resolved reliably, and are consistent with the findings from patch-clamp measurements that POPG decreases the rate of desensitization. Of note, the time constants of desensitization of ELIC in 2:1:1 POPC:POPE:POPG membranes were significantly slower in the stopped-flow flux assay ($\tau \sim 9$ s) compared to patch-clamp recordings ($\tau \sim 2$ s). This is likely because the concentration of cysteamine used in the flux assay was significantly lower (5 mM) than in the patch-clamp measurements (30 mM). A lower cysteamine concentration was necessary in the flux assay to avoid precipitation in the presence of thallium. In addition, differences in membrane

potential or membrane curvature (liposome versus excised patch) may have also contributed to this discrepancy. Nevertheless, when comparing differences in desensitization within each assay, the results substantiate the role of POPG in stabilizing the open state relative to the desensitized state, and demonstrate the utility of measuring pLGIC activity in liposomes of defined lipid composition using complementary patch-clamp and stopped-flow flux techniques.

While pLGICs are known to be sensitive to their lipid environment, the binding sites that mediate lipid modulation are not well defined. It has been proposed that TM4 is a lipid sensing structure in pLGICs because of its proximity to the lipid membrane and sensitivity to mutagenesis (*Carswell et al., 2015b*; *Tobimatsu et al., 1987*; *Bouzat et al., 1998*; *Li et al., 1992*). Furthermore, crystal structures of GLIC show bound lipids within intrasubunit grooves between TM4-TM1 and TM4-TM3 (*Bocquet et al., 2009*), which have been proposed to be important determinants of channel opening (*Prevost et al., 2012*; *Basak et al., 2017*). Photolabeling studies have also identified intrasubunit neurosteroid binding sites adjacent to TM4 that mediate neurosteroid modulatory effects (*Cheng et al., 2018*; *Chen et al., 2019*). We show that POPG binding at multiple interfacial arginine residues, including R117, R124, R299, and R301, which are localized to the extracellular and intracellular sides of TM4 (*Figure 8*), are likely important in mediating the effect of POPG on ELIC desensitization. Examination of boundary POPG from coarse-grained simulations with high POPG mole% (50%) at 15 μs shows POPG headgroups making contacts with all of these arginine side chains, and illustrates potential binding modes for the acyl chains (*Figure 8B*). For example, boundary POPG with headgroups that interact with R301 or R299 have acyl chains that make contacts with intrasubunit sites along the intracellular side of TM4 (*Figures 8B* and *5B*). R301, which has the largest effect on desensitization when mutated, is conserved among many mammalian pLGICs including GABA$_A$R and nAchR isoforms, and R299 is adjacent to R301 at the bottom of TM4. Mutations in this region of TM4 have profound effects on pLGIC desensitization (*Bouzat et al., 1998*; *Li et al., 1992*; *Domville and Baenziger, 2018*; *Lee et al., 1994*). R117 and R123 are located at the extracellular end of TM4, and boundary POPG with headgroups that interact with these residues have acyl chains that make contacts with intrasubunit sites on both sides of TM4 (*Figures 8B* and *5B*). Sites equivalent to R123 and R117 in GLIC were previously found to be occupied by a phospholipid and docosahexaenoic acid (DHA), respectively (*Prevost et al., 2012*; *Basak et al., 2017*). The polyunsaturated fatty acid, DHA, was found to increase desensitization in GLIC (*Basak et al., 2017*), although it is possible that DHA could instead be stabilizing a pre-activated state in GLIC (*Gielen and Corringer, 2018*). Nevertheless, it is possible that the exact lipid structure occupying these sites results in different effects. Our results raise the hypothesis that lipids with polyunsaturated acyl chains or certain sterols (*Shen et al., 2000*) exert the opposite effect of activating phospholipids by acting as competitive antagonists.

While the lipid composition of *Erwinia chrysanthemi* membranes has not been determined, the membranes of bacteria from the *Erwinia* genus contain lipids common to gram negative bacteria including PE, PG, and lipid A species (*Sohlenkamp and Geiger, 2016*; *Shukla et al., 1978*). Therefore, while the physiologic function of ELIC is not known, it is possible that changes in the content of the anionic phospholipid, PG, could play a role in modulating ELIC desensitization in *Erwinia chrysanthemi*. In eukaryotic pLGICs, PG is not the most abundant anionic phospholipid, and pLGICs such as the nAchR are also sensitive to other anionic phospholipids such as phosphatidylserine and phosphatidic acid (*daCosta et al., 2009*; *daCosta et al., 2004*). Moreover, a recent cryo-EM structure of the GABA$_A$R in a lipid nanodisc showed a lipid density bound to the bottom of TM4, which was modeled as phosphatidylinositol-4,5-bisphosphate (PIP$_2$) (*Laverty et al., 2019*). While the functional role of PIP$_2$ in the GABA$_A$R is not clear, this structure illustrates the possibility that other anionic phospholipids may modulate eukaryotic pLGIC function through direct binding interactions.

In summary, the anionic phospholipid, POPG, decreases desensitization in the pLGIC, ELIC. POPG specifically binds to and stabilizes ELIC by interacting with interfacial arginine residues. Our results strongly suggest that binding of POPG at specific sites modulates receptor desensitization.

## Materials and methods

### Mutagenesis, expression, and purification of ELIC

pET26-MBP-ELIC was a gift from Raimund Dutzler (Addgene plasmid # 39239) and was used for WT ELIC expression and generation of mutants. Site-directed mutagenesis was performed by the standard Quikchange approach, and confirmed by Sanger sequencing (Genewiz, Plainfield, NJ). WT and mutant ELIC were expressed as previously described (*Basak et al., 2017*; *Hilf and Dutzler, 2008*) in OverExpress C43 (DE3) *E. coli* (Lucigen, Middleton, WI). Cultures were grown in Terrific Broth (Sigma, St. Louis, MO) and induced with 0.1 mM IPTG for ~16 h at 18°C. Pelleted cells were resuspended in Buffer A (20 mM Tris pH 7.5, 100 mM NaCl) with complete EDTA-free protease inhibitor (Roche, Indianapolis, IN), and lysed using an Avestin C5 emulsifier at ~15,000 psi. Membranes were collected by ultracentrifugation, resuspended in Buffer A, solubilized in 1% DDM (Anatrace, Maumee, OH), and incubated with amylose resin (New England Biolabs, Ipswich, MA) for 2 h. The resin was washed with 20 bed volumes of Buffer A, 0.02% DDM, 0.5 mM tris(2-carboxyethyl)phosphine (TCEP), and 1 mM EDTA, and eluted with Buffer A, 0.02% DDM, 0.05 mM TCEP, and 40 mM maltose. Eluted protein was digested overnight with HRV-3C protease (Thermo Fisher, Waltham, MA) (10 units per mg ELIC) at 4°C, and injected on a Sephadex 200 10/300 (GE Healthcare Life Sciences, Pittsburgh, PA) size exclusion column in Buffer A, 0.02% DDM.

### Native MS measurements

Native MS analysis was similar to previous descriptions for other membrane proteins (*Gault et al., 2016*). For analysis of ELIC in DDM, 30 µl of purified protein in 0.02% DDM at ~1 mg/ml was buffer exchanged into 200 mM ammonium acetate pH 7.5% and 0.02% DDM using Biospin six gel filtration spin columns (Bio-Rad, Hercules, CA). 2 µl of buffer exchanged ELIC was loaded into a borosilicate capillary emitter (Thermo Scientific, Waltham, MA), and analyzed by static nanospray on a Thermo QExactive EMR mass spectrometer. The following parameters were used to resolve the ELIC pentamer and minimize dissociation into tetramer and monomer: capillary voltage of 1.2 kV, capillary temperature of 200°C, ion transfer optics set with the injection flatapole, inter-flatapole lens, bent flatapole, transfer multiple as 8, 7, 6, 4 V, respectively, resolution 8,750, AGC target $3 \times 10^6$, trap pressure set to maximum, CID 200 V, and CE 100 V. For analysis of ELIC in C10E5, ELIC delipidated by injecting 300 µg onto a Sephadex 200 10/300 column (GE Healthcare) at 0.5 ml/min pre-equilibrated with Buffer A, 10% glycerol, and 0.06% C10E5 (Anatrace). 30 µl aliquots were then buffer exchanged to 100 mM ammonium acetate pH 7.5, 0.06% C10E5 using Biospin six columns, and diluted to 0.2 mg/ml. MS measurements on the QExactive EMR were performed with the parameters listed above except: capillary temperature 100°C, CID 75 V and CE 200 V. For lipid binding measurements, stocks of POPG lipid were prepared at 2× the concentration of POPG being tested in 100 mM ammonium acetate pH 7.5% and 0.06% C10E5. Lipid stocks were then mixed with 0.4 mg/ml ELIC in a 1:1 vol ratio for a final concentration of 1 µM ELIC, and samples were analyzed after >5 min incubation.

MS spectra were deconvoluted using UniDec (*Marty et al., 2015*); deconvolution of spectra with bound lipid was restricted to the 26+ to 22+ charge states (*Figure 1—figure supplement 3*). Peak heights of apo and lipid-bound species were extracted from UniDec, and analyzed by two approaches. The average number of bound lipids was determined by the following relationship:

$$\text{Average number bound lipid} = \frac{\sum_{n=0}^{k} n \cdot I_n}{\sum_{n=0}^{k} I_n} \qquad (1)$$

where n is the number of bound lipids and $I_n$ is the deconvoluted peak height of ELIC with n bound lipids. Peak heights of apo and lipid-bound species were also plotted as mole fraction versus the number of bound lipids (*Figure 1—figure supplement 4*). These data were fit with a binomial binding model, which assumes that there are N sites each with equal affinity, K. The probability, p, that a site is occupied at the concentration of a given lipid, A, is defined as:

$$p = \frac{[A]}{[A] + K} \tag{2}$$

Then, the probability (B) that q sites are occupied out of N total sites is given by the binomial probability function:

$$B(q) = \frac{N!}{q!(N-q)!} p^{N-q}(1-p)^q \tag{3}$$

B(q) was used to determine the mole fraction of each lipid-bound species at a given [A], which was used to fit the native MS data in Excel across all [A] by setting K constant and varying N or vice versa.

## Thermal stability assay

Purified WT ELIC in C10E5 (Buffer A, 0.06% C10E5) was diluted to 1 µM in the absence or presence of various concentrations of phospholipid. Samples were analyzed without and with heating in the absence or presence of phospholipid. Analysis of protein thermal stability was performed by injecting 90 µl of sample on a size exclusion column (Sephadex 200 10/300), and measuring the amplitude of the pentamer peak as previously described (*Hattori et al., 2012*; *Miller et al., 2017*). Heating was performed for 15 min at 32°C, which resulted in a ~85% decrease in the pentamer amplitude compared to 4°C. The stabilizing effect of a phospholipid was quantified as the pentamer amplitude in the presence of phospholipid (heated) divided by control (heated).

## Excised patch-clamp recordings from giant liposomes

ELIC WT and mutants were reconstituted into giant liposomes as previously described with some modifications (*Matulef and Valiyaveetil, 2018*). Three liposome preparations were used in this study: 1) 25% POPG (consists of 50% POPC/25% POPE/25% POPG), 2) 12% POPG (consists of 60% POPC/28% POPE/12% POPG), and 3) 40% POPG (consists of 35% POPC/25% POPE/40% POPG). These liposome compositions were chosen to vary POPG mole% while optimizing lipid mixtures to obtain ideal giant liposomes for patch-clamping. This was achieved by varying POPG mole% and POPC mole% inversely. Condition #1 was used for WT and all mutants, and conditions #2 and #3 were used in WT. Liposomes were prepared by drying 15–20 mg of lipid mixtures in chloroform using $N_2$ in a round bottom flask and then overnight in a vacuum dessicator. Dried lipids were rehydrated at 5 mg/ml in 10 mM MOPS pH 7, and 150 mM NaCl (MOPS buffer), subjected to 10 freeze-thaw cycles, and then small unilamellar liposomes were formed by extrusion using a 400 nm filter (Avanti Lipids, Alabaster, AL) and bath sonication (30 s × 5). 5 mg of liposomes in 1 ml were destabilized by adding DDM to 0.2% and rotating for 1 h at room temperature followed by 0.3–0.5 mg of ELIC WT or mutants at ~4–5 mg/ml and incubation for 30 min. To remove DDM, SM-2 Bio-beads (Bio-Rad) were added in five batches (30, 30, 50, 100, and 100 mg). The first three batches were added each hour along with 1 ml of MOPS buffer to make a final volume of 4 ml while rotating at room temperature. After adding the first 100 mg batch, the proteoliposomes were rotated overnight at 4°C, followed by the last 100 mg the next day for 3 h at room temperature. Proteoliposomes were harvested by ultracentrifugation at 150,000 × *g* for 1 h at 4°C, and the pellet resuspended with 80 µl of MOPS buffer for a lipid concentration of ~50 mg/ml. Giant liposomes were formed by drying 10 µl of proteoliposomes on a glass coverslip in a desiccator for 3–5 h at 4°C followed by rehydration with 60 µl of MOPS buffer overnight at 4°C and at least 2 h at room temperature the next day. Giant liposomes were resuspended by pipetting and then applied to a petri dish with MOPS buffer.

Patch-clamp recordings were performed using borosilicate glass pipettes pulled to ~2–3 MΩ using a P-2000 puller (Sutter instruments, Novato, CA). Pipettes were filled with 10 mM MOPS pH 7, 150 mM NaCl, and 0.5 mM BaCl₂. Excised patches (the orientation of ELIC in the liposomes is not known; therefore, these patches are not defined as outside-out or inside-out) were held at −60 mV, and bath solutions consisted of 10 mM MOPS pH 7, 150 mM NaCl, 0.5 mM BaCl₂, 1 mM dithiothreitol (DTT), and varying concentrations of cysteamine. DTT was added to the bath solution to prevent cysteamine oxidation. Rapid solution exchange was achieved with a three-barreled flowpipe mounted and adjusted by to a SF-77B fast perfusion system (Warner Instrument Corporation, Hamden, CT). Liquid junction current at the open pipette tip demonstrated 10–90% exchange times

of <10 ms. Data were collected at 20 kHz using an Axopatch 200B amplifier (Molecular Devices, San Jose, CA) and a Digidata 1322A (Molecular Devices) with Axopatch software, and a low pass Bessel filter of 10 kHz was applied to the currents. Analysis of currents was performed with Clampfit 10.4.2 (Molecular Devices). Activation currents were fit to a single exponential equation, and desensitization currents were fit to both single and double exponential equations. The majority of desensitization currents were best fit with a double exponential, and weighted time constants were derived using the following calculation:

$$\text{Weighted Tau} = \frac{(A1 \cdot \tau1) + (A2 \cdot \tau2)}{A1 + A2} \tag{4}$$

where A1 and A2 are the weighted coefficients of the first and second exponential components. The reported weighted average time constants are averages of weighted time constants from double exponential fits and time constants from single exponential fits. Peak cysteamine dose response curves were fit to a Hill equation, keeping n constant at 2, which provided a reasonable fit for all data sets.

## Stopped-flow fluorescence recordings

The fluorescence-based sequential-mixing stopped-flow assay was carried out with an SX20 stopped-flow spectrofluorometer (Applied Photophysics, Leatherhead, UK) at 25°C. To reconstitute ELIC into large unilamellar vesicles (LUVs), 15 mg of lipids (POPC or POPC:POPE:POPG 2:1:1) were dried in glass vials to a thin film under a constant $N_2$ stream. Lipids were further dried under vacuum overnight. The next day, lipids were rehydrated in reconstitution buffer (1114 µl of 15 mM Hepes, 150 mM $NaNO_3$, pH 7). 33 mg CHAPS was added stepwise while sonicating lipids in a bath sonicator until the solution was clear. 1057 µl of a 75 mM ANTS stock solution (in $ddH_2O$, pH 7) was added together with purified ELIC (1 µg/mg lipid), mixed and incubated for 20 min. Detergent removal was initiated by addition of 0.7 g SM-2 BioBeads (BioRad) in assay buffer (10 mM Hepes, 140 mM $NaNO_3$, pH 7). The reconstitution mix was incubated for 2.5 h at 21°C under gentle agitation. The liposome-containing supernatant was transferred to a new glass tube and stored overnight at 13°C. The liposome solution was sonicated in a bath sonicator for 30 s and extruded through a 0.1 µm membrane (Whatman) using a mini-extruder (Avanti Polar lipids). Extra-vesicular ANTS were removed with a 10 ml desalting column (PD-10, GE Lifesciences). Right before the assay, liposomes were diluted 5-fold in assay buffer to ensure a good signal to noise ratio.

For the assay, ELIC-containing liposomes were mixed 1:1 with pre-mix buffer (assay buffer supplemented with 10 mM cysteamine to reach 5 mM after mixing) and incubated for defined amounts of time (10 ms to 25 s). A second 1:1 mixing step was performed with quenching buffer (10 mM Hepes, 90 mM $NaNO_3$, 50 mM $TlNO_3$, pH 7). ANTS fluorescence was excited at 360 nm and the integral fluorescence above 420 nm was recorded for 1 s. For each delay time, at least eight repeats under identical conditions were performed.

To analyze the data, each repeat was visually inspected and outliers were removed. Each remaining repeat was then fitted to a stretched exponential (*Equation 5*) and the rate of Tl+ influx was determined at 2 ms (*Equation 6*).

$$F_t = F_\infty + (F_0 - F_\infty) \cdot e^{\left\{-\left(\frac{t}{\tau}\right)^\beta\right\}} \tag{5}$$

$$k_t = \left(\frac{\beta}{\tau}\right) \cdot \left(\frac{2\,\text{ms}}{\tau}\right)^{(\beta-1)} \tag{6}$$

with $F_t$, $F_\infty$, $F_0$ being the fluorescence at time t, the final fluorescence and the initial fluorescence, respectively. t is the time (in s), $\tau$ the time constant (in s), and $\beta$ the stretched exponential factor. $k_t$ is the calculated rate (in $s^{-1}$) of thallium influx at 2 ms.

The rate constants were averaged and the mean and standard deviations were determined and plotted (*Figure 3F*). The experiments were repeated for each lipid composition using three independent reconstitutions. The rates and standard deviations were averaged and plotted as function of the delay time. The time course was fit according to a double-exponential function:

$$\mathrm{K}_t = A1 \cdot e^{\left\{-\left(\frac{t}{\tau 1}\right)\right\}} + A2 \cdot e^{\left\{-\left(\frac{t}{\tau 2}\right)\right\}} \tag{7}$$

where $\mathrm{K}_t$ is the average rate of thallium influx at a given delay time, A1 and A2 are the amplitudes of each exponential, and τ1 and τ2 are the time constants. The shorter time constant described activation and the longer time constant desensitization.

## Lipid extraction and MS analysis

Lipids were extracted using a Bligh-Dyer extraction (*Bligh and Dyer, 1959*). Briefly, 100 µg of purified ELIC in DDM and 150 µg of *E. coli* membranes derived from cell cultures transformed and induced for ELIC expression, respectively, were mixed with 1 ml chloroform, 2 ml methanol, and 0.8 ml water, and vortexed for 1 min, followed by an additional 1 ml chloroform and 1 ml water, and vortex for 3 min. The samples were centrifuged for 3 min at 500 × *g*, and the lower organic phase removed for analysis, using a Thermo Scientific LTQ Orbitrap Velos mass spectrometer. Lipid extracts were loop injected (1.5 µl/min) using a syringe pump that delivered a continuous flow of methanol at 15 µl/min into the ESI source. High resolution (R = 100,000 at m/z 400) MS and MS/MS analyses were performed in negative ion mode. The skimmer of the ESI source was set at ground potential, electrospray voltage 4 kV, capillary temperature 300°C, AGC target $5 \times 10^4$, and maximum injection time 50 ms. $MS^n$ experiments for identification of lipid structures were carried out with an optimized relative collision energy of 32%, activation q value of 0.25, activation time of 10 ms, and mass selection window of 1 Da.

## Coarse-grained simulations of ELIC

All simulations reported here used the MARTINI 2.2 (*de Jong et al., 2013*) coarse-grained topology and force field. The crystal structure of ELIC (PDB 3RQW) (*Pan et al., 2012*) was coarse-grained using MARTINI martinize.py script. Secondary structural restraints were constructed using martinize.py while imposed through Gromacs (*Van Der Spoel et al., 2005*). Conformational restraints were preserved through harmonic bonds between backbone beads < 0.5 nm apart with a coefficient of 900 kJ mol$^{-1}$. Pairs were determined using the ElNeDyn algorithm (*Periole et al., 2009*). Membranes were constructed using the MARTINI script insane.py (*de Jong et al., 2013*). The insane.py script randomly places lipids throughout both inner and outer membranes and embeds selected proteins into the membrane. Two series of simulations were developed, the first using POPE and POPG, and the second POPC and POPG. Box sizes were about $30 \times 30 \times 25$ nm$^3$ and each simulation box contained about 3000 lipids.

Molecular dynamics simulations were carried out using GROMACS 5.1.4 (*Van Der Spoel et al., 2005*). All systems were run using van der Waals (vdW) and electrostatics in cutoff and reaction-field, respectively, with a dielectric constant of $\varepsilon = 15$. vdW and electrostatics used a cutoff length of 1.1 nm as defined in current MARTINI build specifications. Energy minimizations were performed for about 30,000 steps. All systems were run for short equilibration steps. Canonical ensembles (NVT) were run for 100 ps using a Berendsen thermostat set to 323 K with the temperature coupling constant set to 1 ps. Isothermal-Isobaric ensemble (NPT) equilibration was run for 5000 ps using a Berendsen thermostat and barostat. The thermostat was set to 323 K with the temperature coupling constant set to 1 ps, and the barostat was set to a pressure coupling constant of 3 ps with a compressibility of $3 \times 10^{-5}$ bar$^{-1}$ holding at 1 bar. Molecular dynamics were carried out using NPT ensemble and were simulated for 15 µs with a time step of 0.015 ps using a v-rescale thermostat set to 323 K and a temperature coupling constant of 1 ps. Membranes consisting of POPE used the Parrinello-Rahman barostat, and membranes consisting of POPC used the Berendsen barostat, both under semi-isotropic coupling. The reference pressure was set to 1 bar, the compressibility $3 \times 10^{-4}$ bar$^{-1}$, and the pressure coupling constant 1 ps.

Annular lipids were determined using the annular lipid metric B:

$$B_i = \left\langle \frac{b_i}{b_{tot}} \right\rangle \frac{1}{x_i} - 1 \tag{8}$$

where $b_i$ is the instantaneous number of boundary lipids of species $i$, $b_{tot}$ is the instantaneous total number of boundary lipids, $x_i$ is the overall (bulk) fraction of species $i$ and the brackets represent an average over time and replicas. $B_i < 0$ and $B_i > 0$ indicate enrichment and depletion of species $i$,

respectively, relative to the abundance in the bulk membrane. A given lipid was counted as a boundary lipid if it was within 6 Å of the ELIC transmembrane domain.

Two dimensional lipid density distributions around a central ELIC pentamer were calculated for each leaflet using polar coordinates (*Sharp et al., 1861*). For every sampled frame, all lipids of species $i$ were separated into leaflets. For all $i$ lipids in a given leaflet, the vector separating the phosphate beads from ELIC center was calculated and projected onto the membrane plane. The two-dimensional separation vector was then used to assign the lipid to the appropriate polar bin of radial bin width 4 and angular bin width $\frac{\pi}{15}$. The area density in each bin was averaged over time and replicas.

## Statistical analyses

Statistical comparisons for (1) ELIC channel properties in liposomes with varying %POPG and (2) the native MS POPG binding for WT and mutants were made using a one-way ANOVA with post-hoc Tukey HSD test. Statistical comparisons for ELIC channel properties between WT and mutants were made using simultaneous comparison of pairs (WT vs mutants) with a Bonferroni method to infer statistical significance. All experimental replicates for patch-clamping were technical replicates utilizing different patches from the same liposome preparations, which were stored in the freezer prior to sample preparation and analysis.

## Acknowledgements

We are grateful to Alex Evers, Joe Henry Steinbach, Christopher Lingle, and Gustav Akk for helpful discussions and edits regarding this study and the preparation of the manuscript. We also acknowledge Arthur Laganowsky and Yang Liu for guidance with regard to sample preparation of ELIC for native MS measurements. We are indebted to Michael Gross at the Washington University NIH/NIGMS-supported biomedical mass spectrometry resource for use of the Thermo QExactive EMR mass spectrometer, and Christopher Lingle for use of a patch-clamp rig for electrophysiology recordings. Computational resources were provided through the Rutgers Discovery Informatics Institute.

## Additional information

### Funding

| Funder | Grant reference number | Author |
| --- | --- | --- |
| National Institute of General Medical Sciences | K08GM126336 | Wayland WL Cheng |
| Center for the Investigation of Membrane Excitability Diseases | Pilot Research Grant | Wayland WL Cheng |
| American Heart Association | 18POST33960309 | Philipp AM Schmidpeter |
| National Institute of General Medical Sciences | R01GM124451 | Crina M Nimigean |
| National Institute of Diabetes and Digestive and Kidney Diseases | P30DK020579 | Fong-Fu Hsu |
| National Institute of General Medical Sciences | P41GM103422 | Fong-Fu Hsu |

The National Institute of General Medical Sciences (K08GM126336) and Center for the Investigation of Membrane Excitability Diseases provided the majority of support for this study including funding for study design, data collection and analysis, and the decision to submit the manuscript for publication. The American Heart Association (18POST33960309) and National Institute of General Medical Sciences (R01GM124451) supported the stopped-flow fluorescence recordings.

## Author contributions
Ailing Tong, Philipp AM Schmidpeter, Data curation, Formal analysis, Writing—review and editing; John T Petroff II, Data curation, Writing—review and editing; Fong-Fu Hsu, Data curation, Formal analysis, Methodology; Crina M Nimigean, Methodology, Writing—review and editing; Liam Sharp, Data curation, Formal analysis; Grace Brannigan, Investigation, Methodology, Writing—review and editing; Wayland WL Cheng, Conceptualization, Resources, Data curation, Formal analysis, Supervision, Funding acquisition, Investigation, Methodology, Writing—original draft, Project administration, Writing—review and editing

## Author ORCIDs
Fong-Fu Hsu https://orcid.org/0000-0001-5368-0183
Philipp AM Schmidpeter http://orcid.org/0000-0003-2871-9706
Crina M Nimigean http://orcid.org/0000-0002-6254-4447
Liam Sharp https://orcid.org/0000-0003-3653-949X
Wayland WL Cheng https://orcid.org/0000-0002-9529-9820

## Decision letter and Author response
Decision letter https://doi.org/10.7554/eLife.50766.023
Author response https://doi.org/10.7554/eLife.50766.024

## Additional files

### Supplementary files
• Supplementary file 1. Phophatidylethanolamine and phosphatidylglycerol species identified in lipid extracts by MS/MS. Table shows m/z, intensity, mass, and mass accuracy of each phospholipid species.
DOI: https://doi.org/10.7554/eLife.50766.020
• Transparent reporting form DOI: https://doi.org/10.7554/eLife.50766.021

### Data availability
All data generated and analyzed in this study are included in the manuscript and supporting data.

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
