## [Decision Letter]

Thank you for submitting your article "Direct binding of phosphatidylglycerol at specific sites modulates desensitization of a ligand-gated ion channel" for consideration by *eLife*. Your article has been reviewed by three peer reviewers, including Cynthia Czajkowski as the Reviewing Editor, and the evaluation has been overseen by Richard Aldrich as the Senior Editor. The following individual involved in review of your submission has agreed to reveal their identity: Carol V Robinson.

The reviewers have discussed the reviews with one another and the Reviewing Editor has drafted this decision to help you prepare a revised submission.

Summary:

Using a combination of native mass spectrometry (MS), coarse-grain MD simulations and targeted mutagenesis combined with functional assays, this manuscript examines whether phospholipids directly bind to the bacterial pentameric ligand-gated ion channel (pLGIC), ELIC and can modulate its gating. The reviewers agreed that this is an interesting and well-executed study.

The major results and conclusions are:

1) Native MS data demonstrate that ELIC preferentially binds anionic POPG.

2) Coarse grain MD simulations support the native MS data and indicate that ELIC binds POPG preferentially over POPE and POPC.

3) Patch-clamp electrophysiology of ELIC reconstituted into liposomes of defined lipid composition demonstrate that increasing POPG concentration decreases current macroscopic desensitization. Fluorescent flux assays support the electrophysiological data.

4) Targeted mutagenesis identify 5 arginine residues that when mutated to glutamine decrease POPG binding (native MS) and increase macroscopic desensitization (patch-clamping). MD simulations show POPG binding near the identified arginine residues.

The authors' conclusions that POPG directly binds to ELIC and can modulate channel desensitization kinetics is well supported by the experimental data. The mass spectrometry experiments have been well-executed and together with the electrophysiology, site directed mutagenesis and control experiments with other lipids, the results are compelling. There are many strengths to this manuscript: the manuscript is well written and organized, data are high quality and support the conclusions, multiple experimental approaches were used to support conclusions. The data make significant contributions to our understanding of how lipids regulate pLGIC function, which will have implications for the effects of lipid drugs and other modulators.

Addressing the following essential revisions will strengthen the manuscript.

Essential revisions:

1) Please comment on differences in the time constants for desensitization obtained from patch-clamping versus flux assay. Table 1: 25% POPG: tau 1.95 sec compared to Table 2 POPC:POPE:POPG (2:1:1) tau 0.9 sec. Authors may want the data in Table 2 report tau's versus rate constants so the data reported are consistent throughout the manuscript.

2) Table 1 – Are the changes in desensitization tau's significantly different from each other? No stats are shown.

3) Figure 3E, 3F and Materials and methods – The authors need to provide additional information/description about how rate of desensitization was obtained from the flux assays. Please include some fluorescence traces associated with desensitization since the change in rate of desensitization is the effect that the authors are focused on. Based on patch-clamp electrophysiology data, one would expect over 70% of the channels to be desensitized following a 5 sec exposure to cysteamine, data in Figure 3E do not appear to show any differences in quenching rate at 5 sec compared to 0.1 sec exposure. To highlight the differences in desensitization rates in PC versus PC/PE/PG, including a plot with the data normalized might be helpful.

4) Add some discussion regarding the possible role of lipid sensitivity in ELIC's native environment: what type of lipids might this protein encounter in its prokaryotic host, and what biological purpose might its apparent sensitivity serve? How eukaryotic homologs might require different sensitivities – would provide important biological context.

5) Missing was any discussion of the effects of charge on these interactions. In the positive ion mode of the mass spectrometer, positively charged ELIC binds tightly to negatively charged POPG. What happens when the protein is negatively charged? While this may appear to be a technical point, it is relevant to any discussion about in vivo gating since pH may play a role in regulating opening and closing of a channel (see Liko et al., 2018). The authors should comment on this point.

---

## [Author Response]

Essential revisions:1) Please comment on differences in the time constants for desensitization obtained from patch-clamping versus flux assay. Table 1: 25% POPG: tau 1.95 sec compared to Table 2 POPC:POPE:POPG (2:1:1) tau 0.9 sec. Authors may want the data in Table 2 report tau's versus rate constants so the data reported are consistent throughout the manuscript.

The time constants for desensitization between patch-clamping and liposome flux measurements are indeed different, and we have changed the data in Table 2 to report tau’s so as to maintain consistency in the reported data. This is most likely because the concentration of cysteamine used in the stopped flow assay (5 mM) is 6x lower than in the patch-clamp measurements (30 mM). The use of this lower concentration was necessary because precipitation was observed when mixing 30 mM cysteamine with thallium. Other factors that may contribute to this discrepancy include differences in membrane potential or membrane curvature (liposome vs. excised patch). We have added material in the Discussion section to comment on this.

2) Table 1 – Are the changes in desensitization tau's significantly different from each other? No stats are shown.

The numerical values displayed in Table 1 are also shown as graphs in Figure 3D and Figure 3—figure supplement 2. These graphs display the statistical difference between values, which are difficult to graphically convey in this table since the statistical analyses involves multiple comparisons.

3) Figure 3E, 3F and Materials and methods – The authors need to provide additional information/description about how rate of desensitization was obtained from the flux assays. Please include some fluorescence traces associated with desensitization since the change in rate of desensitization is the effect that the authors are focused on. Based on patch-clamp electrophysiology data, one would expect over 70% of the channels to be desensitized following a 5 sec exposure to cysteamine, data in Figure 3E do not appear to show any differences in quenching rate at 5 sec compared to 0.1 sec exposure. To highlight the differences in desensitization rates in PC versus PC/PE/PG, including a plot with the data normalized might be helpful.

We have added content in the Materials and methods section to describe the analysis of desensitization rate in the flux assay. In addition, we modified Figure 3E by adding a trace for the PC/PE/PG condition with 25 sec exposure to cysteamine, and separating the PC and PC/PE/PG data into separate graphs with different scales so as to better illustrate the difference in desensitization rates between these two conditions. Please view the comments in (1) regarding the difference in desensitization rate between the flux assay and patch-clamp measurements.

4) Add some discussion regarding the possible role of lipid sensitivity in ELIC's native environment: what type of lipids might this protein encounter in its prokaryotic host, and what biological purpose might its apparent sensitivity serve? How eukaryotic homologs might require different sensitivities – would provide important biological context.

We have added content to the Discussion section on the potential role of anionic phospholipid in modulating ELIC in its native environment and the implications of this sensitivity in eukaryotic homologs. While the lipid composition of *Erwinia chrysanthemi* has not been determined and the functional role of ELIC in its host is not known, the lipid membrane composition of bacteria from the *Erwinia* genus contain lipids common to gram negative bacteria including phosphatidylethanolamine (PE), phosphatidylglycerol (PG), and lipid A species (1). *Erwinia carotovora* membranes contain a notably high PE content of nearly 95% (2). Therefore, it is possible that changes in the anionic phospholipid, PG, on the order of 5-10% could fine-tune the function of ELIC in *Erwinia chrysanthemi*. In eurkaryotic homologs, PG is not an abundant anionic phospholipid, and pLGICs such as the nAchR have evolved to be sensitive to other anionic phospholipids such as phosphatidylserine and phosphatidic acid (3, 4). A recent cryo-EM structure of the GABA_A_R in a lipid nanodisc showed a co-purified lipid density modeled as PIP_2_ (5). While the precise functional effect of this bound anionic phospholipid was not determined, this is suggestive that anionic phospholipids also modulate eukaryotic pLGIC function through direct binding interactions.

5) Missing was any discussion of the effects of charge on these interactions. In the positive ion mode of the mass spectrometer, positively charged ELIC binds tightly to negatively charged POPG. What happens when the protein is negatively charged? While this may appear to be a technical point, it is relevant to any discussion about in vivo gating since pH may play a role in regulating opening and closing of a channel (see Liko et al., 2018). The authors should comment on this point.

The reviewers raise an important point regarding the role of charge and positive vs. negative ion mode in POPG binding. We did not perform analyses of lipid binding using negative ion mode. Given the role of positively charged arginine residues in mediating POPG binding, we anticipate that POPG binding affinity in a negatively charged ELIC would be reduced. Interestingly, it has been shown that ELIC channel opening is significantly increased when lowering the pH from 7.4 to 6.5 (6). Thus, it is possible that titration of the protonation of certain amino acids, particularly basic residues, may alter PG binding, which may account for pH dependent modulatory effects on ELIC function. While it is beyond the scope of this study to test this possibility, we have added a section in our Discussion to comment on this point.

References:

1) Sohlenkamp C, Geiger O. Bacterial membrane lipids: diversity in structures and pathways. FEMS Microbiol Rev. 2016;40(1):133-59. doi: 10.1093/femsre/fuv008. PubMed PMID: 25862689.

2) Shukla SD, Green C, Turner JM. Proteins, phospholipid distribution and fluidity in membranes of the Gram-negative bacterium *Erwinia carotovora* [proceedings]. Biochem Soc Trans. 1978;6(6):1347-9. doi: 10.1042/bst0061347. PubMed PMID: 744424.

3) daCosta CJ, Medaglia SA, Lavigne N, Wang S, Carswell CL, Baenziger JE. Anionic lipids allosterically modulate multiple nicotinic acetylcholine receptor conformational equilibria. J Biol Chem. 2009;284(49):33841-9. doi: 10.1074/jbc.M109.048280. PubMed PMID: 19815550; PMCID: PMC2797154.

4) daCosta CJ, Wagg ID, McKay ME, Baenziger JE. Phosphatidic acid and phosphatidylserine have distinct structural and functional interactions with the nicotinic acetylcholine receptor. J Biol Chem. 2004;279(15):14967-74. doi: 10.1074/jbc.M310037200. PubMed PMID: 14752108.

5.) Laverty D, Desai R, Uchanski T, Masiulis S, Stec WJ, Malinauskas T, Zivanov J, Pardon E, Steyaert J, Miller KW, Aricescu AR. Cryo-EM structure of the human alpha1beta3gamma2 GABA_A_ receptor in a lipid bilayer. Nature. 2019;565(7740):516-20. doi: 10.1038/s41586-018-0833-4. PubMed PMID: 30602789; PMCID: PMC6364807.

6.) onzalez-Gutierrez G, Lukk T, Agarwal V, Papke D, Nair SK, Grosman C. Mutations that stabilize the open state of the *Erwinia chrisanthemi* ligand-gated ion channel fail to change the conformation of the pore domain in crystals. Proc Natl Acad Sci U S A. 2012;109(16):6331-6. doi: 10.1073/pnas.1119268109. PubMed PMID: 22474383; PMCID: PMC3341056.